# Replacing the Burden of the Gluten Free Diet: Then, Now, and the Future

**DOI:** 10.3390/ijms232315108

**Published:** 2022-12-01

**Authors:** Roxana Nemteanu, Irina Ciortescu, Corina Elena Hincu, Andreea Clim, Liliana Gheorghe, Anca Trifan, Alina Plesa

**Affiliations:** 1Medical I Department, “Grigore T. Popa” University of Medicine and Pharmacy, 700115 Iasi, Romania; 2Institute of Gastroenterology and Hepatology, “Sfantul. Spiridon” University Hospital, 700111 Iasi, Romania; 3Department of Radiology, “Sfantul Spiridon” University Hospital, 700111 Iasi, Romania

**Keywords:** celiac disease, treatment, gluten-free

## Abstract

Without a doubt, a majority of diseases are food-pattern-related. However, one disease stands out as an increasingly more common autoimmune-mediated enteropathy triggered by the ingestion of gluten. Celiac disease (CD) is an old disease, with changing clinical patterns, affecting any age, including infancy and adolescence, and becoming more frequent among the elderly. The gluten-free diet (GFD) has been the sole provider of clinical, serological, and histological improvement for patients with CD for more than seven decades. Nowadays, complete avoidance of dietary gluten is rarely possible because of the wide availability of wheat and other processed foods that contain even more gluten, to the detriment of gluten-free products. Undeniably, there is a definite need for replacing the burdensome GFD. An add-on therapy that could control the dietary transgressions and inadvertent gluten consumption that can possibly lead to overt CD should be considered while on GFD. Nevertheless, future drugs should be able to provide patients some freedom to self-manage CD and increase food independence, while actively reducing exposure and mucosal damage and alleviating GI symptoms. Numerous clinical trials assessing different molecules have already been performed with favorable outcomes, and hopefully they will soon be available for patient use.

## 1. Introduction

The 21st century has become the age of social media, for better and for worse. The digital voice is a powerful tool that could help reduce the burden of chronic disorders, major contributors to human morbidity and mortality, by reaching all individuals, regardless of age and social background [1]. Currently, cardiovascular and cerebrovascular diseases, cancer, diabetes, and hypertension have reached epidemic proportions, putting a financial strain on the healthcare system [2]. Inflammatory bowel diseases (IBD) have emerged with difficult-to-treat disease patterns, constantly challenging patients and healthcare providers. New insight into the subtle link between metabolic and liver diseases, the increase of obesity-related disorders, nonalcoholic steatohepatitis, and liver cancers have emerged in recent years masking a silent but dangerous enemy [3]. The consumption of inexpensive, high-calorie, nutrient-poor fast foods with high fats, salts, and sugars has been increasing throughout the developing world, affecting all social groups [4]. Constant efforts are being directed to developing medical devices, drug discovery, and other pharmacological advances to reduce this burden which could be more easily managed by the promotion and early implementation of healthier dietary habits [5].

Without a doubt, a majority of diseases are food-pattern-related. Therefore, the “naturals revolution” which we are witnessing today does not come as a surprise [5,6]. Food-oriented companies have directed their efforts toward creating natural functional and fortified food products as a response to the consumer’s increased attention toward more conscious choices [6]. This global resurgence in interest has shifted the interest of the scientific community of modern medicine towards finding new natural bioactive molecules derived from herbs, plants, and foods, and assessing their potential role as therapeutic and also prophylactic agents, as a viable alternative to pharmaceutical drugs suspected of having side effects and adverse reactions [7]. This is just the tip of the iceberg in modulating disease occurrence and phenotype, changing patterns and outcomes.

However, one disease stands out as an increasingly more common, autoimmune-mediated enteropathy, occurring in the genetically susceptible and triggered by the ingestion of gluten proteins from wheat (named gliadin and glutenin), barley (hordeins), and rye (secalin) [8,9]. It is an old disease, with changing clinical patterns, affecting any age, including infancy and adolescence, and becoming more frequent among the elderly [10]. Currently, 1.4% of the world’s population has CD, with an increasing incidence of around 7.5% each year, which is why CD is considered an important health problem [11]. We are not certain that the consumption of gluten itself plays a role in growing incidence rates, but we can state that the wide spread of accurate diagnostic tests and increased awareness among general practitioners and other medical physicians have led to a better screening and diagnostic process [12]. As previously reported, CD is the result of a combination of genetic and environmental factors that enable the loss of tolerance to gluten proteins and progression toward overt CD while favoring additional food intolerance [13]. The gut microbiome and the metabolome, together with an inappropriate immune response, have been the object of clinical research, trying to find the missing links in disease pathogenesis and new therapies for patients with CD.

In this manuscript, we aimed to review the current state of development of several new treatment options for patients with CD. In light of more recent discoveries, we performed a literature search between January 2012 and October 2022 on MEDLINE, EMBASE, Web of Science, Scopus, DDW.org, and ClinicalTrials.gov for English articles. The search terms used included, but were not limited to, “celiac disease”, “therapy”, “therapeutic”, “therapeutic options”, “new”, “novel”, “alternative”, “advances”, and “drug therapy”. The cited articles were selected based on their relevance to the review objective.

## 2. Genetics and Genes in CD

Genetic susceptibility to CD is represented by predisposing HLA-DQ2 or HLA-DQ8 haplotypes. These haplotypes are found in approximately 30–40% of the general population and although necessary for disease occurrence, they are not sufficient [14]. Regional variants have been observed in the frequency of these haplotypes in CD; however, their distribution is ubiquitous [15]. Exposure to gluten, the known CD exogenous antigen, is mandatory, but the quantity, quality, and timing of exposure are still a matter of debate. The identification of additional predisposing genes was obtained from the Genome-Wide Association Studies performed on populations of different geographical backgrounds [16]. These studies identified 39 loci associated with CD development and other autoimmune diseases [17]. Single nucleotide polymorphisms (SNPs) analysis showed that some loci are associated with more than one CD gene, raising the number of polymorphisms to 57 [17,18]. Cerquieira et al. identified the TLR7/TLR8 locus associated with CD onset before seven years of age, whereas the SH2B3/ATXN2, ITGA4/UBE2E3, and IL2/IL21 loci were associated with later development of CD and a more severe mucosal injury [19]. It seems that their role is to regulate gene expressions via chromatin status, resulting in epigenetic modifications, a new field that is under evaluation [20,21]. In this domain, categories such as new RNA sequencing and microRNA studies, DNA methylations, and histone modifications resulting in chromatin remodeling, are still under assessment as possible [20,21]. However, these advanced studies have proven that the identified loci were not able to completely explain CD genetic predisposition since HLA plus all these additional genes accounted for about 54% of the hereditability of CD [22,23]. The remaining 46% of genetic vulnerability is representant by non-HLA genes which have minor roles [24].

## 3. Mechanisms Involved in CD Occurrence: Gluten Peptides, Cytokines, and Immune-Mediated Responses

Gluten is a complex protein found in grains. Gluten composition varies between both species as well as cultivars [25]. Structurally, gluten is composed of two major proteins, glutenin and gliadin [25,26]. Because of their high glutamine and proline content, gluten proteins are incompletely degraded by digestive enzymes in the human stomach, resulting in the presence of peptides (most importantly from gliadin) that contain immunogenic epitopes (Table 1). Gluten is classified as high molecular weight (HMW) and low molecular weight (LMW) glutenins; the S-poor ω-gliadins; and S-rich α, β, and γ-gliadins [26]. Difficulties in proteolysis are encountered in the normal intestine, also. Consequently, there is an accumulation of peptide fragments in the gut lumen that gain access to the lamina propria, either actively by transepithelial translocation or passively by paracellular flux through a compromised epithelial barrier [27].

Intestinal permeability is another factor that plays a role in the development of CD, because of the increased transport of undigested gluten particles from the lumen to the lamina propria, and a “leaky gut” might trigger the early stages of innate immune activation [28]. Intestinal permeability and tight junction (TJ) functionality are influenced by a human protein modulator called Zonulin. It can reversibly regulate the paracellular migration of protease-resistant gluten fragments [29]. It is upregulated due to gluten exposure and, as a result, the undigested peptides pass through the enterocyte barrier and enter lamina propria where they are deaminated by the enzyme tissue transglutaminase 2 (TTG2) [8,30]. Inhibiting the activity of Zonulin and decreasing intestinal permeability has been a promising field for the development of new therapies.

CD engages both innate and adaptive immune branches of the host’s immune system in response to proteolytically resistant gluten peptides [31]. When reaching the lamina propria, these active peptides encounter a key player in the physiological process, the TTG2, a calcium-dependent transamidation enzyme produced by cells during inflammation [32]. Since its discovery by Dieterich et al. in 1997 as the most important autoantigen targeted by autoantibodies, this 76 kD enzyme has allowed researchers to significantly increase the body of knowledge regarding CD pathophysiology and has offered insight into advanced therapeutic options [33]. TTG2 is expressed by all cell types, but it is stored in an inactive enzymatic form in the extracellular matrix, inside the cell, and on the cell surface [34]. The TTG2 selectively deamidates the positively charged glutamines into negatively charged glutamic acid residues [35]. Then, the deamidated gliadin peptides together with TTG2 complexes are endocytosed by antigen-presenting cells (APC) represented by dendritic cells or macrophages [36]. The negatively charged surface increases its bonding affinity for the HLA-DQ2 or HLA-DQ8 heterodimers situated on APC. The gluten epitopes are recognized by the HLA DQ molecules and then further activate gluten-specific CD4+ T cells via T cell receptor (TCR) recognition [37].

APCs can recognize tissue distress via the release of interleukin-15 (IL-15) and type I interferon (IFN) and enable the differentiation of gluten-specific HLA-DQ2- or HLA-DQ8-restricted CD4+ T cells [31,38]. Once stimulated, these cells produce pro-inflammatory cytokines and also engage TTG2 and gluten-specific B cells, thereby encouraging their activation and differentiation into antibody-producing cells (IgA and IgG anti-TTG2 and anti-gliadin antibodies) [34,38]. The cytokine release via activation of Th1 (activates CD8+ T cells and NK cells) and Th2 (stimulates B cells into antibody-secreting plasma cells) inflammatory pathways cause cell death adding to the injury [39,40]. In addition, activated IELs express tumor necrosis factor-alpha (TNF-a) in the mucosa. With anti-TNF agents available in IBD treatment, we cannot but wonder if there is a place for them in CD treatment. However, a paper by Turner et al. showed that patients with active uncomplicated CD compared to refractory CD (RCD) would not benefit from anti-TNF-a agents such as infliximab [41]. Cytotoxicity and decreased apoptosis of IELs are a consequence of increased expression of IL-15 in enterocytes [42]. It is important to emphasize the relationship and cross-talk between innate immune cells and epithelial cells in CD patients in the early stages of the disease. This type of communication, along with changes in the patient’s innate immune system, could explain the loss of gluten tolerance and also constitute the foundation for developing new drugs for CD patients.

## 4. Out with the Old Gluten-Free Diet?

The gluten-free diet has been the sole provider of clinical, serological, and histological improvement for patients with CD for more than seven decades [43]. Nowadays, complete avoidance of dietary gluten is rarely possible because of the wide availability of wheat and other processed foods that contain even more gluten to the detriment of gluten-free products [42]. Patients face numerous challenges in maintaining an adequate diet because of the financial burden of specific GF products, which are more expensive, rarely subsidized by the government, and not always available in stores [42]. It is difficult to assess correctly the amount of gluten a patient consumes daily. Also, the tolerance threshold is unique for each individual, and their reactiveness and immune response are uniquely linked to environmental and microbiome changes. In addition, any deflection from the strict GFD, which is limited to a maximum of 20 mg/kg per day, may result in ongoing silent mucosal damage. As a result, compliance with the GFD is inconsistent, and patients follow a gluten-reduced diet rather than a strict GFD [44]. However, despite all odds, the GFD continues to gain popularity beyond its primary role of managing patients with gluten-induced immune-mediated disorders. It is expected to restore the histology of the bowel architecture in more than 90% of children within two years of diet, whereas up to two thirds of adult patients experience mucosal recovery after two to five years of GFD [10,34]. With small bowel recovery, a GFD can also improve the signs and symptoms of malabsorption and the QoL for CD patients, while reducing the overall disease burden [34,43]. In addition, it is argued that the GFD may help improve neurological impairment seen in gluten ataxia, depression, the autism spectrum disorder, and “foggy brain” [43]. Monitoring gluten exposure using fecal and urine samples to measure GIP concentrations is a useful and noninvasive tool to assess compliance, but unfortunately these tests are not widely available [45]. Ruiz-Carnicer et al. determined the GIP concentrations in three consecutive urine samples collected within a week and confirmed that 58% of patients under GFD had recent gluten intake. Moreover, the GIP levels showed a negative predictive value of 97% regarding the presence of histological lesions [46]. Silvester et al. noticed that single urine or stool determination of GIP would be inaccurate to state gluten exposure because the excretion kinetics of gluten are highly variable among individuals [47]. Recently, point of care tests (POCTs) for CD have been developed and are commercially available. POCTs seek to bring testing and disease monitoring closer to the patients. Baseline patient assessment using POCTs allows fast results and therefore supports the efficacy of clinical decision-making [20,21]. A quick and reliable diagnosis of CD may be obtained after using a POCT, and more importantly these easy-to-use tests may be included in follow-up protocols. POCTs may help streamline the follow-up process by providing TTG2 results during the consultation and enable the decision making regarding the necessity of follow-up duodenal biopsy [8,9]. Follow-up strategies are encouraged in current guidelines, and usually include a combination of tests (serological, biochemical), biopsy sampling, and symptom control [13]. DEXA, pneumococcal vaccination, and psychological evaluation are also recommended during follow-up [20,34,40]. Nutritional education, patient support groups, and dietary support should be offered to patients with CD from day one of diagnosis. Patients with CD should be referred to a dedicated dietitian who is well-trained concerning CD in order to get a detailed nutritional assessment, education on the GFD, and subsequent monitoring [12,25]. A dietetic review supported by questionnaires is a useful tool to provide education for a balanced and adequate but not excessive nutrient intake [20,40]. Therefore, a clear need for other types of therapy has emerged and numerous pathways have been explored. New drugs are currently under evaluation in clinical trials, but their efficiency is still to be determined (Table 2).

## 5. Less Aging More Healing: Is There a Role for Probiotics and Polyphenols in CD?

Gut microbiota plays a key role in maintaining intestinal homeostasis and promoting health. Recent research has linked the role of the microbiome to the loss of gluten tolerance and increased recruitment of T lymphocytes [11]. Francavilla et al. assessed the efficacy and safety of a probiotic mixture in patients with CD. It was reported that a 6-week course of a mixture of five strains of lactic acid bacteria and bifidobacteria (*Lactobacillus casei*, *Lactobacillus plantarum*, *Bifidobacterium animalis* subsp. *lactis* Bi1, *Bifidobacterium breve* Bbr8, and *B. breve* Bl10) was effective in improving the severity of IBS-type symptoms in CD patients on a strict GFD [48]. Similar results were reported in the pediatric population by Basit Ali et al. Probiotics were found to be highly efficient in terms of reduction in diarrhea in CD and are useful tools to be considered in improving the quality of life among young patients with CD [49]. However, the data reported are contradictory. A recent meta-analysis by Selier et al. found no significant difference regarding improvement of GI symptoms or QoL compared to placebo [50].

On the other hand, recent reports have suggested that a continuous and prolonged intake of fruits, vegetables, and whole grains—all rich sources of polyphenols—may promote intestinal health in patients with CD. Therefore, polyphenols may become valuable tools against the cytotoxicity of gluten prolamins, reduce oxidative stress and inflammation, optimize intestinal epithelial barrier integrity and function, and modulate immune responses [1]. However, the exact role of polyphenols in the management of CD patients is still to be determined.

## 6. New Insight on CD Treatment

New trends in the research development of novel CD therapies have focused mainly on finding drugs able to interfere with different pathways in disease pathogenesis. Therefore, blocking or interfering with gluten absorption, TTG2 activity, and neutralizing HLADQ2/DQ8 epitopes are currently of interest [31].

The induction of gliadin-specific immune tolerance is a promising therapeutic target for a wide range of diseases such as allergic, autoimmune, or post-transplant organ rejection [51]. Therefore, inhibition of specific immune responses by inducing a non-responsiveness antigenic state towards unknown or self-antigens has been extensively studied [51,52]. Kelly et al. assessed the efficiency and tolerance of TAK-101, formerly known as TIMP-GLIA, gliadin encapsulated in poly nanoparticles. The randomized, double-blind, proof-of-concept study was conducted assessing plasma nanoparticle-free gliadin concentrations [53]. TAK-101 was administered via a 30-min intravenous infusion of 0.1–8 mg/kg (maximum of 650 mg), followed by dose escalation according to the study protocol. The two-phase clinical trial aimed to assess as the primary endpoint the safety and tolerability of TAK-101 administered intravenously in patients with CD, and reduction in mucosal damage, measured as the change from baseline in villus height and change in intraepithelial lymphocyte (IELs) density [53]. TAK 101 demonstrated tolerogenic inhibition of T-cell activation and possible reduction in the deterioration of intestinal villi following gluten challenge [53].

## 7. Microbial Therapeutic Enzymes: A Promising Area of Biopharmaceuticals

The search for exogenous sources of proteolytic enzymes capable of degrading gluten-derived peptides is an attractive ongoing process. Numerous enzymes linked to their ability to degrade gluten-derived peptides found in bacteria, fungi, and plants have been studied with conflicting results [54]. These enzymes are known as glutenases and are subdivided depending on their catalytic mechanism. The majority of these endopeptidases are serine, cysteine, aspartic, and metalloendopeptidases [55]. Among serine endopeptidases, prolyl endopeptidases (PEP) are capable of effectively hydrolyzing peptides smaller than 30 residues [56]. PEPs can specifically break the peptide bond of proline residues at the carboxyl end. The catalytic triad responsible for the catalytic properties is comprised of Ser, His, and Asp, and has a large β-propeller domain and a small N-terminal catalytic domain [54,55,56].

Latiglutenase (IMGX-003), a combination of ALV001 and ALV002, a cysteine endopeptidase B-isoform 2 that is derived from barley (EP-B2), and a *Sphingomonas capsulate* PEP (SC-PEP), respectively, showed promising initial results [57]. The combination of the two enzymes is complimentary. It is thought that EP-B2 efficiently digests the 33-mer peptides into smaller, not necessarily non-toxic proline-containing fragments, and PEP then digests the proline–glutamine links in these smaller oligopeptides [58]. However, a reduction in enzymatic activity in the gastric environment was reported. The preliminary results of clinical trials showed that latiglutenase failed to improve symptom and histological scores. One of the major concerns was the possibility of immunotoxic residues reaching the duodenum if gastric emptying occurred before digestion. Latiglutenase is capable of degrading only small amounts of gluten and it was hypothesized that it could be used for managing inadvertent contamination for CD patients [57,58]. However, Xiao et al. recognized the hidden potential of this enzyme and, using computer-aided rational protein design tools, identified mutants and assessed experimentally their activities. The engineering tools used showed significant results of an increase of 80–200% of the catalytic rate [59]. It was found that a conformational transition of the B-propeller domain and catalytic domain of PEP, a shorter distance between the substrate and the oxyanion holes, were crucial for enzymatic activity [59]. The research could be the basis for implementing this technology for future use in designing and creating highly active SC-PEP mutants effective for gluten degradation [59].

Other sources, including fungal enzymes such as AN-PEP which exhibited post-proline cleavage activity, were also considered as potential therapeutic agents for CD [57,58]. AN-PEP showed efficiency in degrading gliadin fractions and also HMW and LMW glutenin. The commercially available product known as Tolerase G, a food supplement, is currently available for the CD population [58]. While Tolerase G effectively reduced the amount of consumed gluten that was exposed to the duodenum in a clinical study, it did not completely degrade the gluten [57,60]. Therefore, it is not an effective treatment for CD, as no safe concentration of gluten in the duodenum has been determined, and the same study even states that AN-PEP is not intended to treat or prevent CD [60,61,62]. Tolerase G, being a dietary supplement, is also not FDA regulated [61,62]. Nevertheless, numerous scientists have reported contradictory results for AN-PEP efficiency. Tack et al. assessed during a randomized, double-blind, placebo-controlled pilot study the different outcomes for CD patients consuming gluten with AN-PEP for 2 weeks, and after a 2-week washout period randomly assigned to gluten intake with either AN-PEP or placebo [60]. The authors concluded that AN-PEP appeared to be safe in CD patients, and it was able to reduce small bowel IgA-TTG2 deposits compared to placebo, but further studies with larger CD populations and longer timeframe are required [60].

Further research highlighted the importance of two commercially available food-grade proteases, aspergillopepsin from *Aspergillus niger* and dipeptidyl peptidase IV derived from *Aspergillus oryzae* (STAN 1) [57,63]. *Aspergillus niger* is found in different environments, especially in decomposing plants, and seldom as a human pathogen [57,63,64]. *Aspergillus oryzae*, used for oriental food fermentation, is currently used for its ability to produce enzymes including proteases and cellulases [63,64]. Benoit-Gelber et al. showed that *A. niger* and *A. oryzae* mixed cultures in wheat bran secreted a broader range of degrading enzymes than respective monocultures [64]. Ehren et al. assessed the detox properties in a randomized clinical trial of patients with CD receiving gluten and placebo or this protease mixture and found unsatisfactory results in regard to TTG IgA antibodies levels [65]. Although these enzymes are widely used in food processing, STAN1 did not produce sufficient results to further consider it as a viable treatment option for CD in its current state.

However, TAK-062, an endopeptidase engineered from Kuma030, the bacterial enzyme kumamolisin-As from *Alicyclobacillus sendaiensis*, was assessed by Pultz et al. in a phase I clinical trial involving CD patients and healthy subjects [66]. The safety, tolerability, and efficacy of TAK-062 were evaluated. The results showed that close to 99% of gluten was degraded by TAK-062 300 mg in a liquid or capsulated formulation, according to the study protocol. TAK-062 is capable of breaking down large amounts of gluten, up to 9 g within physiologically relevant time scales, assuring a high degradation rate both in vitro and in the stomach [66]. The drug was tested in patients treated with PPIs and some concerns were expressed because of the altering of the pH level, but pretreatment did not affect the enzyme’s capacity to degrade gluten [66]. The highest tested dose of 2000 mg was safe, well tolerated, and without adverse side effects. No significant differences were noted regarding delivery by liquid or capsules, and no systemic exposure was noted. The specificity of TAK-062 for the P-Q motif in gliadin is the reason for this therapeutic success, and a phase II of the clinical trials is to be expected [66].

## 8. The Leaky Gut: More Than Meets the Intestinal Barrier

The transcellular or paracellular trafficking of gliadin peptides is dependent on TJ integrity. The delivery of gliadin peptides to lamina propria is conditioned by intestinal permeability and TJ access. Zonulin is a key player in the pathogenesis of CD and can be used as a biomarker of impaired gut barrier function [67]. Zonulin causes TJ disassembly, thereby leading to increased intestinal permeability irrespective of disease activity. Thus, zonulin antagonists have been proposed and evaluated as a possible therapeutic approach to prevent TJ disassembly, thereby restoring epithelial barrier integrity [29,67]. AT-1001 (larazotide acetate) is structurally similar to an isolated toxin generated by the *Vibrio cholera* bacterium and has been extensively studied as an option for CD patients [29,67]. In a meta-analysis and review by Hoilat et al., a total of 626 patients receiving larazotide acetate and placebo were assessed. AT-1001 is largely well tolerated and superior to placebo in reducing GI symptoms among CD patients undergoing gluten challenge [68]. Leffler et al. in a multicenter, randomized, double-blind, placebo-controlled study assessed the potential of larazotide acetate administered three times per day in different doses (0.5, 1, or 2 mg) to relieve ongoing symptoms in 342 adults with CD who had been on a GFD for ≥12 months. Larazotide acetate in association with the GFD reduced signs and symptoms in CD patients better than a GFD alone [69]. Larazotide acetate is one of the very few drugs that entered phase III fast-track trials and is on its way to FDA approval.

## 9. Vaccines and CD

Vaccines induce a state of immune tolerance and non-reactivity towards different antigens. Therefore, the possibility of using a vaccine for treating CD patients has emerged. Nexvax2 is a novel, peptide-based, epitope-specific immunotherapy that contains three gluten-derived peptides, intended to tolerize CD patients to gluten [70]. Clinical research states that Nexvax2’s induction of non-responsiveness and reactivity disappears after repeated doses, or is avoided with gradual dose escalation [70,71]. The downside is that the vaccine is only suitable for patients with the HLA-DQ2 haplotype (the majority of patients with CD). A separate vaccine would therefore have to be investigated for HLA-DQ8-positive patients [72]. The safety and tolerability of Nexvax2 in participants with CD on a GFD evaluated in randomized, double-blind, placebo-controlled clinical trials by stepwise dose escalation followed by a high maintenance dose were thoroughly assessed [71]. There are still some challenges with Nexvax2 and future studies will hopefully establish if a vaccine is a suitable treatment option for CD patients [70,71,72].

## 10. Blocking Tissue Transglutaminase, HLA DQ2/DQ8 Molecules, and IL-15

The deamidation process and the production of deamidated immunogenic gluten peptides are performed by TTG2, which increases binding affinity with the DQ2/DQ8 dendritic cells [30,33,34]. Therefore, blocking the activity of either the TTG2 enzyme or the HLA DQ2 and DQ8 molecules has been an attractive research field [33,34]. Inhibition of gliadin peptide deamidation using TTG2 inhibitors reduces the binding affinity for APC [38,40]. However, the ubiquitous TTG2 enzyme has different functions, primarily in wound repair and healing by regulating the activity of different cell types recruited by the damaged tissues [33,34].

TTG2 inhibitors can be either competitive or irreversible inhibitors, competing for active sites or irreversibly binding to TTG2 with loss of enzymatic function [58]. The inhibitors known as ZED1098, ZED1219, and ZED1227 show specificities for the intestinal TTG2 [73]. Among them, ZED1227 completed phase 1 studies and is under phase 2 of clinical trials. However, the safety profile of TTG2 inhibitors is still under investigation because of the side effects reported among mouse models [57,73].

HLA DQ2 and DQ8 blockers are peptides that have been engineered by amino acid substitution and dimerization or introduction of aldehyde groups and have shown moderate efficacy by inhibiting IFN production [57,74]. However, these peptides have shown partial agonist effects on gliadin-stimulated T-cells, causing an augmented immune response [57,74]. Therefore, studies investigating HLADQ2 inhibition are currently being conducted to identify highly specific molecules, high binding affinity, and non-toxic and non-immunogenic antagonists to assess their efficacy and usefulness [57].

The IL-15 molecule is a pro-inflammatory cytokine responsible for CD occurrence and progression to RCD or EATL [42]. Anti-IL-15 monoclonal antibodies have also been studied and promising results were reported, showing a potential role in the reversal of intestinal damage [75]. AMG714 (currently known as PRV 15) has been already tested among patients with rheumatoid arthritis with excellent results [76]. Results from clinical trials involving CD and AMG714 are still to be published.

## 11. Role of Anti-TNF, Anti-IFN γ, and Inhibition of Integrin α4β7

Recent data published by Shah et al. are consistent with the notion that CD is a risk factor for IBD and that, to a lesser degree, patients with IBD have an increased risk of CD [77,78]. Similar results were reported in a systematic review and meta-analysis by Pinto-Sanchez. The authors found that there is likely a bidirectional association between CD and IBD, with a nine-fold increased risk of IBD in CD compared with controls, and a higher risk in Crohn’s disease than UC [77]. This close relationship could be explained by common genetic, immunological, and environmental factors at play. Therefore, it is not surprising that therapeutic agents used for managing IBD patients are slowly transitioning toward the CD population [78].

Vedolizumab (VDZ) is the first licensed, gut-selective biological agent used to treat IBD [79]. VDZ is a humanized immunoglobulin (Ig) G1 monoclonal antibody which binds to α4β7 [80]. However, as promising as it may be, the results of clinical trials regarding the efficiency of VDZ in the management of CD patients are yet to be released [63,79].

Published case reports have shown that infliximab is an effective treatment that may be considered in a small number of patients with refractory CD and resistant to other therapy [81]. Other molecules such as itolizumab and certolizumab were also assessed. For example, itolizumab, an anti-CD6 humanized IgG1 mAb, binds to domain-1 of CD-6 that is responsible for priming, activation, and differentiation of T-cells [8]. Itolizumab significantly reduces T-cell proliferation along with substantial downregulation of the production of cytokines [82]. It has been used for treating patients with psoriasis, but its role in CD is to be determined.

## 12. Conclusions

The GFD is and will remain the mainstay of treatment available for CD patients until safer and more effective alternatives are available. However, there is a definite need for replacing the burdensome GFD. An add-on therapy that could control the dietary transgressions and inadvertent gluten consumption that can possibly lead to overt CD should be considered while on a GFD. Unfortunately, clinical trials on emerging new drugs have reported only partial results, and whilst the GFD still poses some challenges, it represents the best and safest therapeutic choice. It has been proven time and time again that the GFD is sufficient to reduce mucosal damage and improve the patient’s QoL, criteria yet to be fulfilled by novel drugs. Nevertheless, future drugs should be able to provide patients some freedom to self-manage CD and increase food independence, while actively reducing exposure and mucosal damage and alleviating GI symptoms. Numerous clinical trials assessing different molecules have already been performed with favorable outcomes, and hopefully they will be soon available for patient use.

## Figures and Tables

**Table 1 ijms-23-15108-t001:** Classification of different gluten immunogenic peptides and relevant T-cell epitopes.

Cereal Type		Gluten-Derived Peptides	Epitopes
WHEAT	prolamins	α-gliadins γ-gliadins ω-1,2,5-gliadins	DQ2.5, DQ8, DQ8.5, DQ2.2 DQ2.5, DQ8, DQ8.5, DQ2.2 DQ2.5
	glutamines	HMW-glutenins LMW-glutenins	DQ8, DQ8.5 DQ2.5, DQ2.2
RYE	prolamins	γ-secalins ω-secalins	DQ2.5
	glutamines	HMW-secalins	DQ2.5
BARLEY	prolamins	C-hordeins γ/B-hordeins	DQ2.5
	glutamines	D-hordeins B/γ-hordeins	DQ2.5

**Table 2 ijms-23-15108-t002:** Short description of emerging new drugs for CD patients, their structure, role, and partial results from clinical trials.

Treatment Class	Name of Agent	Structure	Role	The Current Stage of Clinical Trials and Expected Therapeutic Outcome
Pretreated flower	Pre-treated flowers with probiotics	*Lactobacillus* *Bifidobacterium infantis* *Bifidobacterium natrum*	*Lactobacillus* strains have enzymatic abilities for hydrolyzing gluten peptides. Protects enterocytes from gliadin-induced damage.	No guideline recommendations for role in symptom management.
Genetic modification of grains	Wheat deletion lines - ω-, γ-gliadins, and LMW-glutenins on short arm of chromosome 1D - α-gliadins on short arm of chromosome 6D	wheat variants with decreased immunogenicity via genetic engineering	Confer less immunogenicity and also have lower proportions of αβγω gliadins.	Preclinical phase, no data is available.
Co-polymeric binders	BL 7010	a synthetic, nonabsorbable copolymer of styrene sulfate with hydroxyethyl methacrylate	Has a high affinity to α-gliadin peptides. It retains intraluminal gliadin and prevents splitting into immunogenic peptides.	The result is to be published; concerns about safety profile and binding with other medications.
Enzymatic gluten hydrolysis via endopeptidase	Latiglutenase (formerly ALV003) IMGX-003	a combination of 2 enzymes ALV001 and ALV002	ALV001 degrades gluten proteins and reduces the immunotoxic potential. ALV002 catalyzes the post-proline hydrolysis.	Latiglutenase inhibits gluten from crossing the intestinal barrier. It may be effective in reducing CD symptoms but does not induce mucosal healing.
	STAN1	a combination of microbial enzymes	Degradation of gluten before absorption in the GI tract, expected to decrease persistently elevated TTG levels.	Randomized, placebo-controlled, double-blinded clinical trial; disappointing results, therapeutic role unclear.
	AN-PEP	a second PEP derived from the fungus *Aspergillus niger*	Degradation of gluten.	Randomized, placebo-controlled study of adult patients with CD on a GFD. It did not prevent mucosal injury.
	TAK-101	an immune-modifying nanoparticle	It contains gliadin protein encapsulated in negatively charged poly designed to induce gluten-specific tolerance.	Randomized, double-blind, placebo-controlled. Promotes T-cell suppression by binding to inflammatory cells. Reduces the immune response to antigens.
	TAK-062	a highly potent, computationally designed, third-generation enzyme derived from Kuma030, the bacterial enzyme kumamolisin-As, from Alicyclobacillus sendaiensis	Increased proteolytic activity, and resistance to the gastric and intestinal pepsin and trypsin. The high substrate specificity of TAK-062 is expected to result in a high level of gliadin degradation in the stomach, irrespective of meal composition.	TAK-062 is well tolerated and has demonstrated high specificity and potency in the human stomach. TAK-062 offers potential as an oral therapeutic for the treatment of CD.
Tight junction blockade via zonulin inhibition	Larazotid acetat (AT 1001)	synthetic octapeptide similar in structure to the zonula occludens toxin	Improves TJ integrity and reduces mucosal inflammation.	Larazotide acetate was able to reduce gluten-induced immune reactivity and gastrointestinal symptoms. Good clinical profile.
Gluten vaccine	Immunsan T Nexvax 2	delivery of 16-mer peptides derived from α-gliadin, ω-gliadin, and hordein	Suppress T-cell-mediated inflammation and promote gluten tolerance.	Only suitable for HLA DQ2 genotype. Concerns with a higher risk of autoimmune system activation, causing potential progression of the disease to refractory forms or the development of other autoimmune diseases.
Transglutaminase inhibitors	Competitive, reversible, and irreversible TTG2 inhibitors ZED1098, ZED1219, and ZED1227 Synthetic polymer poly (hydroxymethyl methacrylate-costyrene sulfonate) Anti-gliadin IgY Dihydroisoxazo-les Cinnamoytriazo-le Aryl β-aminoethyl ketones	cystamine is a competitive TTG2 inhibitor that has been evaluated in cultures of duodenal tissue from patients with CD, where it has been found to block the proliferative capacity of T-cells; dihydroisoxazole derivative is an irreversible TTG22 inhibitor which has been studied in rodents	Inhibition of gliadin peptide deamination using TTG2 inhibitors reduces the peptides’ binding affinity for APC.	A new generation of selective inhibitors, this novel therapy is in the early stages of research. ZED1227 completed phase 1 studies and is under phase 2 of clinical trials.
Inhibition of integrin α4β7	Vedolizumab	a humanized monoclonal antibody, which acts against α_4_β_7_ integrin heterodimer and blocks the interaction of α_4_β_7_ integrin with MAdCAM-1	It prevents leukocyte binding to the endothelial surface and its extravasation into the affected tissue.	The efficacy and safety of vedolizumab have been established in many clinical studies. It has shown promising results in various clinical studies.
Anti-IFN-γ- and anti-TNF-therapies	Adalimumab Infliximab Certolizumab Fontolizumab Itolizumab	anti-TNF monoclonal antibodies anti-IFN γ monoclonal antibody	The pro-inflammatory cytokines IFN- and TNF- are important molecules involved in CD pathogenesis secreted by T-cells in response to gluten.	No data available.
The antagonist of IL 15	Anti-IL15 antibodies AMG714	anti-IL 15 monoclonal antibodies	Neutralization of IL 15 could reduce the intestinal injury.	Results from clinical trials involving CD and AMG714 are still to be published.

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
