# Peer review of "Replacing the Burden of the Gluten Free Diet: Then, Now, and the Future"

_ijms, 2022, doi:10.3390/ijms232315108_

Round 1

Reviewer 1 Report

The presented manuscript is a thorough and transparently written review of the mechanism of celiac disease with a focus on management, possible prevention and treatment of the disease. The authors present extensive set of different approaches to therapy, from which the reader can get a clear picture of current situation as well as get a sense of which approaches hold the most promise in the future. The English is error-free and easy to read. I myself have no comments and propose publication in its current form.

Author Response

We thank the reviewer for taking the time to review the manuscript and give positive feedback.

Reviewer 2 Report

The paper by Nemteanu et al entitled “Replacing the burden of the gluten free diet: then, now and the future” is a very interesting text that deals in a very complete way with the alternatives that are being tested to complement the DSG in the treatment of CD. It is publishable, although certain issues could be improved:

-   Although the text contemplates a wide and profuse bibliography, it would be interesting to describe at some point the methodology followed for its selection.

 -          Drugs, vaccines, probiotics, in short, alternatives, are proposed to reduce the stress of the DSG. It is made in an optimistic way, highlighting the positive results of the studies carried out to date, but it would be interesting to lower this degree of speculation and be more objective in relation to direct effects, side effects, possibility of real application in the long term, etc.

For example, in the section on probiotics and polyphenols, a logical benefit of these compounds is presented, which in case of CD could help in the treatment of symptoms in a transversal way, but it is not a specific treatment for the pathology, is it? Therefore, although the title of this section suggests this idea, the text does not make it clear.

Another example could be section 6, where possible alternatives are given in brief but without specifying their possible negative consequences.

In general, it would be interesting to give this more realistic point of view on the different alternatives that are presented. This idea could also be transferred to the last section of the conclusions.

 -          I understand that the aim of the text is to reduce the burden on the GFD, as clearly indicated in the title, I agree. However, the section on the GFD is too strict in relation to the assessment of its benefits, which it is not consistent with the optimism that other therapies are presented (although it is consistent with the idea of providing alternatives to this treatment). GFD is presented as a complicated, expensive, difficult, impossible therapy... which it may be... but it is not sufficiently emphasized that at present it is the only form of treatment for CD.  However, when the other alternatives are presented, it is done in a much less demanding way.

It would be interesting to also provide ways to improve the follow-up of the GFD; tools, such as GIPs, POCT, nutritional education..., should be deepened.

 -          Table 2 lacks of heading

This is a very complete text that integrates all the mentioned alternatives, also well presented using clarifying tables

Author Response

We appreciate the reviewer for taking the time to carefully review the manuscript and give detailed and constructive comments, which has greatly helped to improve this paper. Below is our point-by-point response to each comment.

Reviewer 2: The paper by Nemteanu et al entitled “Replacing the burden of the gluten free diet: then, now and the future” is a very interesting text that deals in a very complete way with the alternatives that are being tested to complement the DSG in the treatment of CD. It is publishable, although certain issues could be improved:

  • Although the text contemplates a wide and profuse bibliography, it would be interesting to describe at some point the methodology followed for its selection.

Thank you for your observation. We have revised the paper and included information regarding the selection process and methodology. Please see page 2 lines 73-80.

  • Drugs, vaccines, probiotics, in short, alternatives, are proposed to reduce the stress of the DSG. It is made in an optimistic way, highlighting the positive results of the studies carried out to date, but it would be interesting to lower this degree of speculation and be more objective in relation to direct effects, side effects, possibility of real application in the long term, etc. For example, in the section on probiotics and polyphenols, a logical benefit of these compounds is presented, which in case of CD could help in the treatment of symptoms in a transversal way, but it is not a specific treatment for the pathology, is it? Therefore, although the title of this section suggests this idea, the text does not make it clear.

Thank you for your constructive remarks. We can agree with the reviewer’s comments. Indeed, although highly debated and unspecific for CD, there is still speculation about the exact role and effects of probiotics and polyphenols in CD. The majority of studies published up to date have reported a positive response after probiotic and/or polyphenol treatment or in the worst-case scenario no clinical or histological improvement for CD patients.

  • Another example could be section 6, where possible alternatives are given in brief but without specifying their possible negative consequences.

Thank you for your comments. The body of information collected concerning newer molecules is still incomplete as clinical trials have released only partial results. The data available, including possible negative outcomes, are briefly summarized in table 2.

  • In general, it would be interesting to give this more realistic point of view on the different alternatives that are presented. This idea could also be transferred to the last section of the conclusions.

Thank you for your comments. We have added information in the Conclusions section. Please see lines 415-419.

  • I understand that the aim of the text is to reduce the burden on the GFD, as clearly indicated in the title, and I agree. However, the section on the GFD is too strict in relation to the assessment of its benefits, which it is not consistent with the optimism that other therapies are presented (although it is consistent with the idea of providing alternatives to this treatment). GFD is presented as a complicated, expensive, difficult, impossible therapy... which it may be... but it is not sufficiently emphasized that at present it is the only form of treatment for CD.  However, when the other alternatives are presented, it is done in a much less demanding way.

Thank you for your observations. We have revised the paper and included information regarding the importance of GFD. Please see page 5-line 181 and page 6 lines 182-189.

  • It would be interesting to also provide ways to improve the follow-up of the GFD; tools, such as GIPs, POCT, nutritional education..., should be deepened.

Thank you for your comments. Please see additional information on page 6 lines 196-211.

  • Table 2 lacks of heading

 Thank you for your observation. We have added the heading, please see page 6 lines 215-216.